# Research on the Dynamic Behaviors of the Jet System of Adaptive Fire-Fighting Monitors

**Xiaoming Yuan [1,2], Xuan Zhu [1], Chu Wang [1,\*], Lijie Zhang [1] and Yong Zhu [3,\*]** 

[1] Hebei Key Laboratory of Heavy Machinery Fluid Power Transmission and Control, Yanshan University, Qinhuangdao 066004, China; yuanxiaoming@ysu.edu.cn (X.Y.); zx00712@126.com (X.Z.); zhangljys@126.com (L.Z.)
[2] State Key Laboratory of Fluid Power & Mechatronic Systems, Zhejiang University, Hangzhou 310027, China
[3] National Research Center of Pumps, Jiangsu University, Zhenjiang 212013, China
[\*] Correspondence: wangchu@stumail.ysu.edu.cn (C.W.); zhuyong@ujs.edu.cn (Y.Z.)

**Abstract:** Based on the principles of nonlinear dynamics, a dynamic model of the jet system for adaptive fire-fighting monitors was established. The influence of nonlinear fluid spring force on the dynamic model was described by the Duffing equation. Results of numerical calculation indicate that the nonlinear action of the fluid spring force leads to the nonlinear dynamic behavior of the jet system and fluid gas content, fluid pressure, excitation frequency, and excitation amplitude are the key factors affecting the dynamics of the jet system. When the excitation frequency is close to the natural frequency of the corresponding linear dynamic system, a sudden change in vibration amplitude occurs. The designed adaptive fire-fighting monitor had no multi-cycle, bifurcation, or chaos in the range of design parameters, which was consistent with the stroboscopic sampling results in the dynamic experiment of jet system. This research can provide a basis for the dynamic design and optimization of the adaptive fire-fighting monitor, and similar equipment.

**Keywords:** adaptive control; fire-fighting monitor; jet system; dynamics; duffing equation; flow control

## 1. Introduction

A fire-fighting monitor is a piece of fire-fighting equipment with a large flow and long-range, and is mainly composed of a barrel and a gun head [1,2]. Through the electromechanical control system of the barrel, horizontal and pitching rotation of the fire-fighting monitor can be realized, so that fires can be extinguished rapidly due to the burning object being sprayed directly with fluid. The gun head is the end effector of the fire-fighting jet system, which converts the pressure energy of the fluid into kinetic energy. The two jet states, spray jet and straight jet, can be switched by adjusting the electromechanical control system of the gun head [2–4]. The nozzle opening of the traditional fire-fighting monitor remains unchanged during the jet flow, and the flow, pressure, and nozzle opening of the jet system are not well matched. Based on the principle of valve components in a hydraulic system [5,6], an adaptive fire-fighting monitor with an elastic adaptive adjustment mechanism at the front end of the nozzle was designed. The nozzle opening of the monitor could be automatically adjusted according to the changes in the flow and pressure of the jet system to improve the performance of the jet system, and its fire extinguishing efficiency, in a wider flow range.

The spray medium of the fire-fighting monitor is generally water, or a mixture of water and foam. During the working process of the fire-fighting jet system, the centrifugal pump, which works as the power source, tends to exhibit pressure pulsation [7–9]. Under the joint action of the pump and the load pressure, the dynamic fluid spring is easily formed due to the compressibility of the

fluid [10,11]. The fluid spring and load mass constitute a dynamic system of the fluid spring and mass The nonlinearity of spring stiffness will cause the natural frequency of the jet system to be non-constant, which will nonlinear vibration of the load mass of the jet system, leading to a loud noise, and even the destruction of the entire structure [12–14]. Similar to the jet system, the valve components in the hydraulic system will also have nonlinear dynamic behaviors such as multi-cycle and chaos under the action of the pressure pulsation, which will cause leakage of the hydraulic valve and other failures [15–17]. Additionally, unstable motion occurs when the railway vehicle reaches high speeds. In the literature [18], the effects of changing parameters with different lateral stiffnesses on nonlinear hunting behavior were analyzed. It was found that the system with bogie and wheelset had less critical speeds than the wheelset system alone, and the increase of the wheelset mass made the hunting behavior even worse. Therefore, it is necessary to explore the effect of a nonlinear fluid spring on the dynamic characteristics of the jet system of adaptive fire-fighting monitors.

At present, the research on the dynamic characteristics of fluid spring systems is generally carried out by methods of analysis, simulation, and experimentation [19–22]. Generally, when using the analytical method, the nonlinear factors of the system are linearized first, and then the system is analyzed by linear control theory, such as the root locus and Bode's chart methods [23–25]. The linearized analysis of nonlinear systems can reveal essential relationships between system parameters and performance evaluation indexes, such as stability, accuracy, and rapidity of the system. However, to some extent, the conclusions are often different from real world observations. It is difficult to explain some of the abnormal phenomena existing in real world dynamic tests, such as complex time-domain waveforms and numerous frequency domain peaks [26]. With the development of nonlinear differential equation solutions, such as the Runge-Kutta algorithm, scholars often use simulation methods to directly calculate the nonlinear system, describing the dynamics of the system more accurately [27–29]. The experimental method is mainly used to verify the results of systematic analysis and simulations, to verify the accuracy of nonlinear system analysis and simulation models [30,31].

In this paper, the adaptive fire-fighting monitor was taken as the research object, and the influence of the nonlinear fluid spring force on the dynamic characteristics of the jet system was the focus of the study. Based on the principles of nonlinear dynamics, a nonlinear dynamic model of the jet system was established. The numerical calculation was used to determine whether the jet system will have dynamic behaviors of deteriorating systems such as multi-cycle, bifurcation, and chaos within the range of operating parameters. Then the measured experimental data was analyzed with nonlinear dynamic research methods, to verify the rationality of the mechanical performance of the designed adaptive fire-fighting monitor.

## 2. Dynamic Modeling of the Jet System

The gun head of the fire-fighting monitor can automatically adjust the nozzle opening with a change of the flow or pressure of the incident fluid, as it can be divided into a traditional diversion gun head and an adaptive gun head. The structure of a traditional diversion gun head is shown in Figure 1. During the working process, except for the outer nozzle which is used for adjusting the state of the jet, the other components are in a relatively static state. Therefore, the nozzle opening cannot be changed with the change of the jet flow and pressure, and an aggravation of turbulent flow and sudden increase of pressure is prone to occur, resulting in a reduction of fire extinguishing efficiency. The structure of an adaptive gun head is shown in Figure 2. Inside the adaptive gun head, an adaptive mechanism is added, consisting of a spray core, an end cap, a core rod, and a spring. When the flow of the incident fluid in the gun head increases, the fluid force on the left side of the spray core increases. When the fluid force is greater than the spring force on the right side of the spray core, the spray core moves to the right, and the nozzle opening increases. In contrast, When the flow of the incident fluid decreases, the spray core moves to the left, and the nozzle opening decreases. Therefore, an adaptive fire-fighting monitor with an adaptive mechanism can automatically adjust the nozzle

opening according to changes in inlet flow and pressure, thereby achieving good jet performance in a wider flow range.

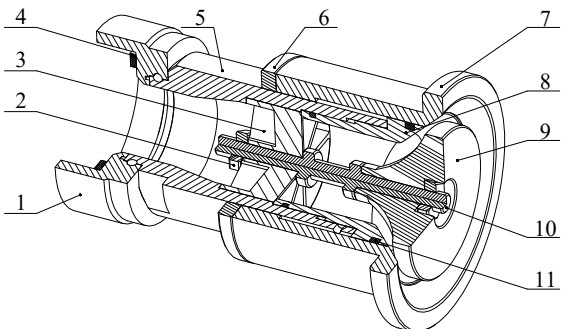

**Figure 1.** Structure of traditional diversion gun head. The labels are as follows: 1. Joint, 2. Nut, 3. Regulator, 4. Gasket, 5. Enclosure, 6. Ring, 7. Outer nozzle, 8. Inner nozzle, 9. Spray core, 10. Endcap, and 11. Core rod.

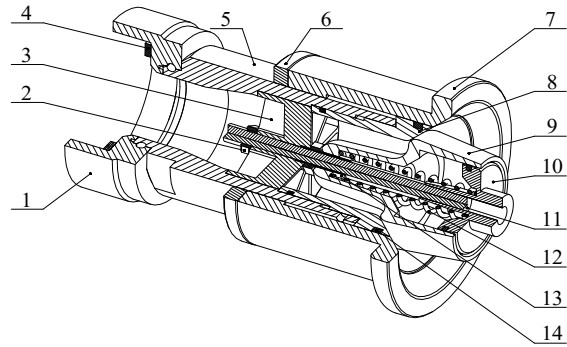

**Figure 2.** Structure of adaptive gun head. The labels are as follows: 1. Joint, 2. Nut, 3. Regulator, 4. Gasket, 5. Enclosure, 6. Ring, 7. Outer nozzle, 8. Inner nozzle, 9. Spray core, 10. Endcap, 11. Core rod, 12. Spring, 13. Core sleeve, and 14. Seal ring.

The connection between the adaptive gun head, the barrel, and the pipeline is shown in Figure 3. The electromechanical system installed on the barrel can realize the horizontal and pitching rotation of the fire-fighting monitor.

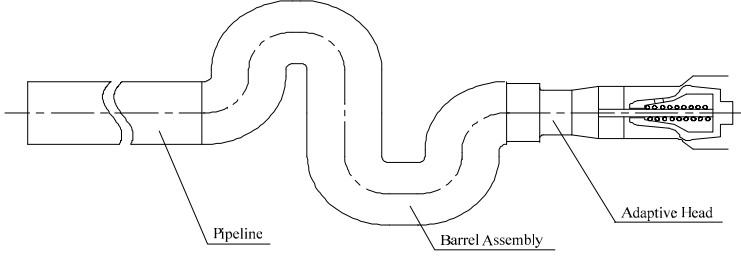

**Figure 3.** The jet system of an adaptive fire-fighting monitor.

The jet system of the adaptive fire-fighting monitor has better jet performance due to the addition of the adaptive mechanism in the gun head. However, the mechanical spring introduced in the mechanism reduces the stiffness of the jet system, and the compressibility of the fluid changes the stiffness of the jet system dynamically, increasing the complexity of the dynamic behaviors of the jet system. When the excitation frequency is close to the natural frequency of the jet system, the vibration of the spray core is intensified, which will significantly reduce the fire extinguishing efficiency. The internal structure of the adaptive gun head is very similar to that of the valve element in a hydraulic

system. Therefore, referring to the dynamic analysis of the relief valve, the working principle of the jet system of an adaptive fire-fighting monitor is shown in Figure 4.

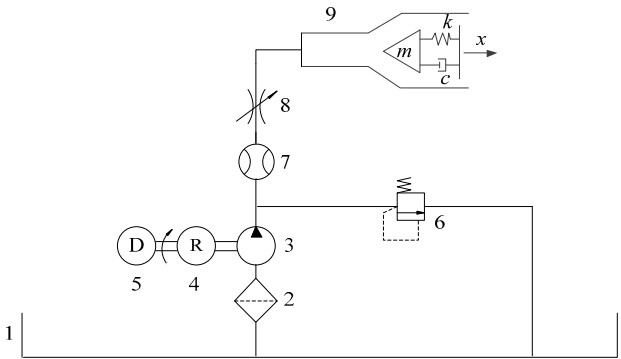

**Figure 4.** Working principle of the jet system of an adaptive fire-fighting monitor. The labels are as follows: 1. Water tank, 2. Filter, 3. Pump, 4. Reducer, 5. Diesel, 6. Relief valve, 7. Flowmeter, 8. Throttle valve, and 9. Adaptive fire-fighting monitor.

It can be seen in Figure 4 that the diesel and the reducer installed on the fire engine drive the main spindle of the pump to rotate, and the water in the water tank is filtered by the filter, and then sucked into the interior by the pump. The water discharged from the pump enters the adaptive fire-fighting monitor through the flowmeter and the throttle valve, and finally shoots at the fire point.

In Figure 4, $m$ is the mass of the spray core. Analyzing the forces of the spray core, the dynamic equation of the spray core is

$$m\ddot{x} + F_c + F_s = F \tag{1}$$

where, $x$ is the displacement of the spray core under the action of jet fluid, $F_c$ is the viscous resistance, $F_s$ is the spring force, and $F$ is the fluid force on the spray core.

The total equivalent stiffness of the jet system, as shown in Figure 4, is made up of the stiffness of the mechanical spring inside the spray core and the stiffness of the fluid unit on the left side of the spray core. During the operation of the jet system, the movement of the spray core causes a change in the length of the fluid unit on the left side, which in turn causes a change in the stiffness of the fluid unit, ultimately resulting in a change in the total spring stiffness of the jet system. Therefore, the variation law of total stiffness is

$$k(x) = \frac{B_f S}{L + x} + k_s \tag{2}$$

where $B_f$ is the bulk elastic modulus of the fluid. Considering the compressibility of the gas-containing fluid, $B_f$ is calculated by the bulk elastic modulus formula [32]. $S$ is the equivalent cross-sectional area of the fluid unit, $L$ is the equivalent length of the fluid unit, and $k_s$ is the stiffness of the mechanical spring inside the spray core.

Let $y$ be the vibration displacement near the working point of the spray core, $x$, that is $y = \Delta x$. According to the Taylor series, the total stiffness of the jet system near the operating point can be expressed as:

$$k(x + y) = k(x) + \dot{k}(x)y + \ddot{k}(x)y^2/2 + o\left(y^2\right) \tag{3}$$

Assuming $k(x) = k_1$, $\dot{k}(x) = k_2$, and $\ddot{k}(x) = k_3$, then substituting them into Equation (3):

$$k(x + y) = k_1 + k_2 y + k_3 y^2 + o\left(y^2\right) \tag{4}$$

Omitting the infinitesimal of higher order $o(y^2)$ in Equation (4), the total stiffness of the jet system can be expressed as:

$$F_s = k(x + y)y = k_1 y + k_2 y^2 + k_3 y^3 \tag{5}$$

Since the elastic potential energy $U$ of the spring has symmetry, the total elastic potential energy of the jet system can be expressed as:

$$U = k_1 y^2 / 2 + k_3 y^4 / 4 \tag{6}$$

The nonlinear spring force of the jet system can be further expressed by Equation (6):

$$F_s = dU/dy = k_1 y + k_3 y^3 \tag{7}$$

This paper mainly studies the influence of nonlinear spring force on the dynamic characteristics of the jet system, so the nonlinear factors such as friction and system damping are not considered. Then the dynamic equation of the jet system near the working point $x$ is

$$m\ddot{y} + c\dot{y} + k_1 y + k_3 y^3 = F_0 \cos(\omega t + \varphi_0) \tag{8}$$

where, $c$ is the linear damping coefficient of the system, which is the sum of the system structural damping coefficient $c_0$, and the fluid damping coefficient $c_1$. $F_0 \cos(\omega t + \varphi_0)$ is the external periodic excitation caused by the pressure pulsation of the incident fluid, $F_0$ is the amplitude of the external excitation, $\omega$ is the angular frequency of external excitation, and $\varphi_0$ is the initial phase angle of external excitation.

In order to analyze the jet system more conveniently and intuitively, the mass unit of Equation (8) can be normalized as

$$\ddot{y} + 2\xi\omega_0\dot{y} + \omega_0^2 y + \beta y^3 = F_1 \cos(\omega t + \varphi_0) \tag{9}$$

where $\xi = \frac{c}{2\sqrt{k_1 m}}$, $\omega_0 = \sqrt{\frac{k_1}{m}}$, $\beta = \sqrt{\frac{k_3}{m}}$, $F_1 = \frac{F_0}{m}$. $\xi$ is the linear damping ratio, $\omega_0$ is the natural frequency of the linear harmonic oscillator when the nonlinear term coefficient $\beta$ is equal to 0, and $F_1$ is the amplitude of external excitation received by the unit mass.

## 3. Dynamic Analysis of the Jet System

It can be seen from Equation (9) that the dynamic model of the jet system of the adaptive fire-fighting monitor can be described by a Duffing equation with damping. Therefore, the basic laws of the jet system can be revealed by the characteristics of the Duffing equation. The design parameters of the jet system are shown in Table 1:

**Table 1.** Design parameters of the jet system.

| Parameter Name | Parameter Name | Unit | Value |
|---|---|---|---|
| Mass of spray core | $m$ | kg | 0.3163 |
| Structural damping coefficient | $c_0$ | N/(m/s) | 0.01 |
| Fluid damping coefficient | $c_1$ | N/(m/s) | 0.1 |
| Equivalent section of fluid unit | $S$ | m$^2$ | 0.0056 |
| Equivalent length of fluid unit | $L$ | m | 1.6679 |
| Displacement of spray core working point | $x$ | mm | 4.284 |
| Stiffness of spring in the gun head | $k_s$ | kN/m | 18 |
| Temperature | $T$ | K | 293 |

According to the theory of bulk elastic modulus of gas-liquid fluid, the gas content of the fluid ($\alpha_0$), and the fluid pressure ($p$), are the main factors affecting the bulk elastic modulus of the fluid. It can be seen from Equation (2) that these two factors also have direct effects on the stiffness of the jet system. Combining the amplitude ($F_0$) and frequency ($\omega$) of the external excitation in Equation (9), the dynamic characteristics of the jet system can be researched with variables $\alpha_0$, $p$, $\omega$, and $F_0$.

### 3.1. Resonance Analysis of the Jet System

When the incident fluid gas content ($\alpha_0$) is 2%, the fluid pressure ($p$) is 0.6 MPa, and the external excitation amplitude ($F_0$) values of 1 N, 3 N, and 5 N, respectively. The bifurcation, with a frequency of external excitation ($\omega$), is shown in Figure 5.

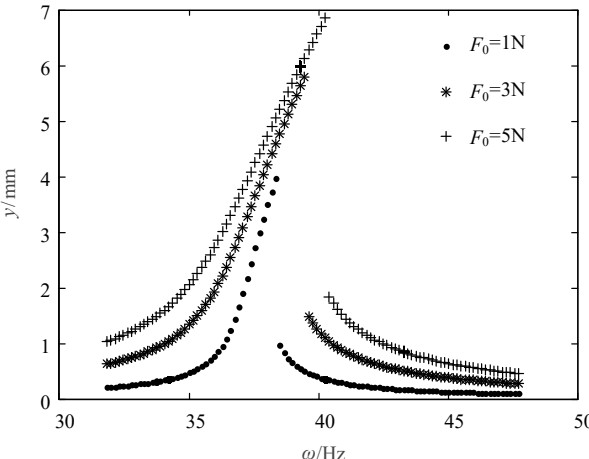

**Figure 5.** Bifurcation of nonlinear dynamic equations of a jet system with a frequency of external excitation ($\omega$).

It can be seen from Figure 5, that when $\omega$ and $F_0$ have different values, the motion of the jet system is a single-cycle vibration. Under the action of different external excitation amplitudes, the amplitude of the jet system first increases and then decreases with the increase of the external excitation frequency. Moreover, the larger the external excitation amplitude, the larger the amplitude of the jet system. There are amplitude mutations in all three curves, and the larger the external excitation amplitude, the higher the external excitation frequency when the amplitude mutation occurs.

Taking $\alpha_0 = 2\%$, $p = 0.6$ MPa, $\omega = 46.8$ Hz, and $F_0 = 3$ N as an example, the Rouge-Kutta method was used to calculate in order to reflect the operating state of the jet system vividly. The time course, the phase diagram, the power spectrum density, and the Poincare map are shown in Figure 6.

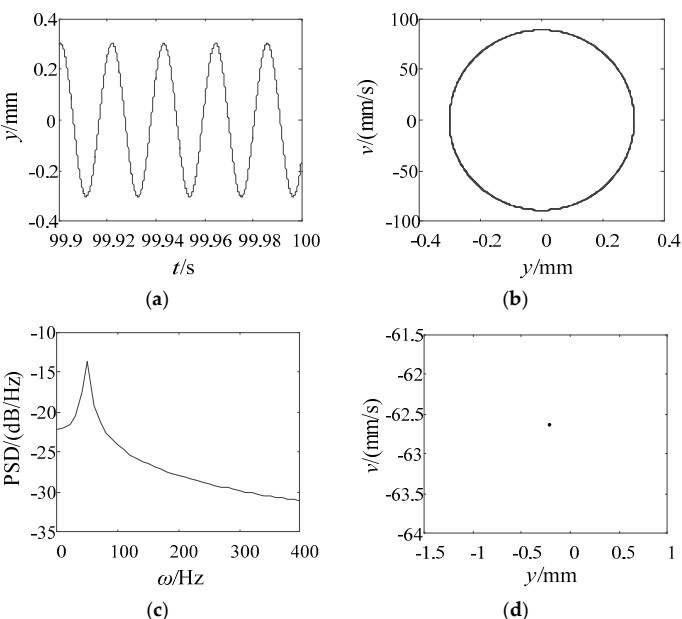

**Figure 6.** Results of dynamics calculation of the jet system. (**a**) Time course, (**b**) Phase diagram, (**c**) Power spectrum density, (**d**) Poincare map.

It can be seen from Figure 6 that the time course is periodically repeated, i.e., the phase diagram is repeated in a finite region, which is a closed curve, that is, there exists a limit cycle; the power spectrum density has a peak at an external excitation frequency of 46.8 Hz; Poincare map has only one point in a certain area. The above are obvious single-cycle vibration characteristics, indicating that when $\alpha_0$ is 2%, $p$ is 0.6 MPa, $\omega$ is 46.8 Hz, and $F_0$ is 3 N, the jet system is in a single-cycle vibration state. The results of the bifurcation of a large number of samples show that when $\alpha_0$ varies from 0 to 5%, $p$ ranges from 0 to 5 MPa, $\omega$ ranges from 0 to 100 Hz, and $F_0$ ranges from 0 to 5 N, the jet system operates steadily in a single-cycle.

### 3.2. Analysis of the Amplitude Mutation of the Jet System

In order to explore the cause of the amplitude mutation of the jet system shown in Figure 5, the amplitude curve of the jet system with the external excitation frequency shown in Figure 7 was plotted with $p$ = 0.6 MPa, $\alpha_0$ = 2%, $F_0$ = 3 N, and the nonlinear coefficient $\beta$ in Equation (9) had values of 0 and 2849.5 N/(mm$^3$·t), respectively. In Figure 7, $y_1$ is the amplitude when $\beta = 2849.5$ N/(mm$^3$·t), and $y_2$ is the amplitude when $\beta = 0$.

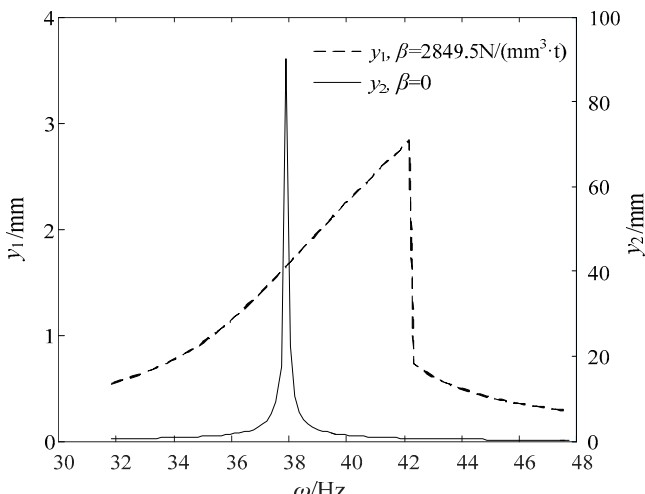

**Figure 7.** Amplitude curve of the jet system with different nonlinear coefficients.

When $\beta = 0$, the dynamic equation of the jet system shown in Equation (9) can be regarded as a linear one. As can be seen from Figure 7, at the natural frequency of about 37.35 Hz, the amplitude reaches its maximum, which is 90.25 mm. Meanwhile, there is no amplitude mutation. In contrast, when $\beta = 2849.5$ N/(mm$^3$·t), the dynamic equation is nonlinear, and the amplitude mutates from 2.833 mm to 0.7359 mm at the external excitation frequency of 42.18 Hz, which is greater than the natural frequency of the linear equation. From the above comparison, it is the nonlinear term that causes the amplitude mutation when the external excitation frequency changes. When the amplitude mutation occurs, the external excitation frequency is greater than the natural frequency of the corresponding linear system, and the maximum amplitude of the jet system is significantly reduced.

### 3.3. Analysis of the Influence of Incident Fluid Pressure and External Excitation Amplitude

When $p$ is 0.6 MPa and $F_0$ is 3 N, and changing $\alpha_0$ and $\omega$, the curve of amplitude of the jet system with $\alpha_0$ and $\omega$ under steady conditions is shown in Figure 8.

It can be seen from Figure 8 that the amplitude mutation occurs in the jet system with an increase of $\omega$. The higher the $\alpha_0$, the lower the $\omega$ and the greater the amplitude, when an amplitude mutation occurs. When $\alpha_0$ is low, the mutation excitation frequency decreases rapidly with the increase of $\alpha_0$. While the rate of decrease of the mutation excitation frequency gradually slows down with the increase

of $\alpha_0$ when $\alpha_0$ is high. When $\omega$ is 38.2 Hz and $\alpha_0$ is 5%, the amplitude reaches the maximum, which is 7.739 mm.

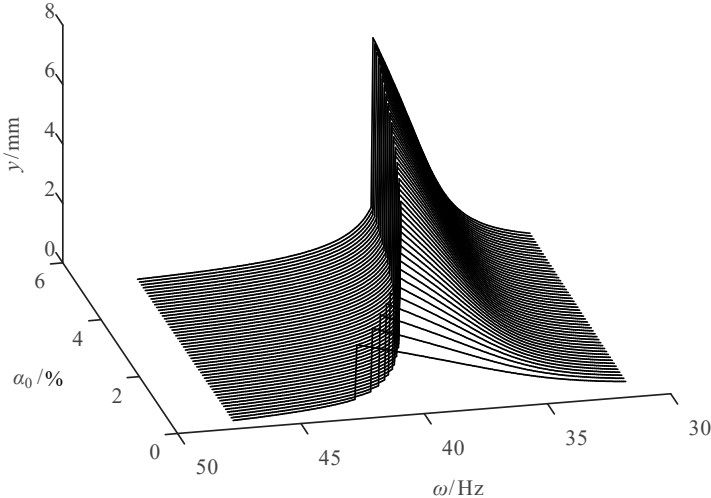

**Figure 8.** Curve of amplitude of the jet system with fluid gas content ($\alpha_0$) and $\omega$.

It can also be seen from Figure 8 that when $\omega$ is less than 38.2 Hz or greater than 42.7 Hz, the amplitude of the jet system changes smoothly with the change of $\alpha_0$. When $\omega$ is between 38.2 Hz and 42.7 Hz, the amplitude of the system has a mutation, showing that the amplitude at this stage is more sensitive to the frequency change.

*3.4. Analysis of the Influence of Incident Fluid Gas Content and External Excitation Amplitude*

When $\alpha_0$ is 2% and $F_0$ is 3 N, changing $p$ and $\omega$, the curve of amplitude of the jet system with $p$ and $\omega$ under steady conditions is shown in Figure 9.

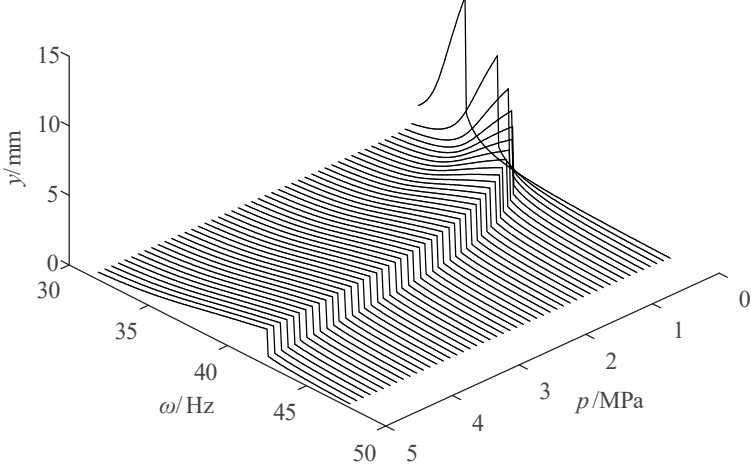

**Figure 9.** Curve of amplitude of the jet system with fluid pressure ($p$) and $\omega$.

It can be seen from Figure 9 that an amplitude mutation occurs in the jet system with an increase of $\omega$. Additionally, the higher the $p$, the higher the $\omega$, and the smaller the amplitude when the amplitude mutation occurs. When $p$ is low, the mutation excitation frequency increases rapidly with the increase in $p$. While the rate of increase of the mutation excitation frequency gradually slows down, with an increase of $p$ when $p$ is high. When $p$ is 0.2 MPa and $\omega$ is 34.78 Hz, the amplitude reaches the maximum, which is 11.48 mm.

It can also be seen from Figure 9 that when $\omega$ is greater than 42.5 Hz, the amplitude of the jet system changes smoothly with the change of $p$. When $\omega$ is between 31.8 Hz and 42.5 Hz, there is a mutation in the amplitude of the system. At this stage, the amplitude was more sensitive to the frequency change, and the sensitive frequencies differed when the pressure is different.

### 3.5. Analysis of the Influence of Incident Fluid Pressure and the Incident Fluid Gas Content

When $\omega$ is 46.8 Hz and $F_0$ is 3 N, changing $p$ and $\alpha_0$, the curve of amplitude of the jet system with $p$ and $\alpha_0$ under steady conditions is shown in Figure 10.

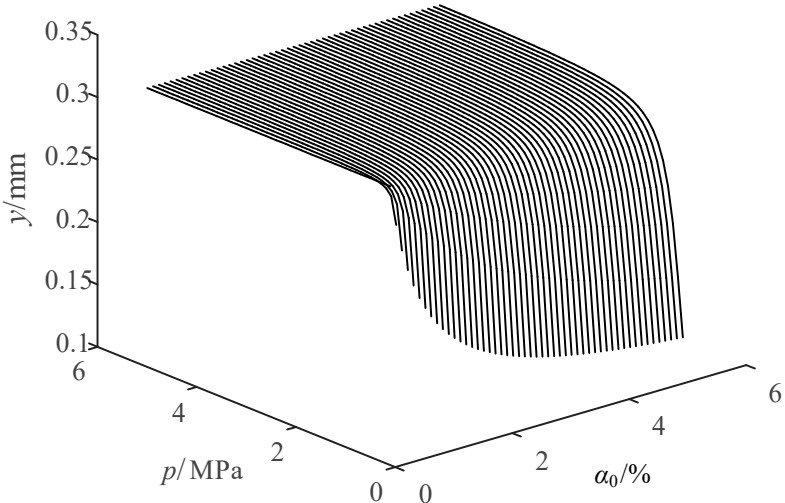

**Figure 10.** Curve of amplitude of the jet system with $p$ and $\alpha_0$.

It can be seen from Figure 10 that when $p$ and $\alpha_0$ change simultaneously, there is no amplitude mutation in the jet system, which is because $\omega$ remains unchanged at 46.8 Hz and is not within the frequency mutation interval, and the interval should also be avoided in the actual jet system to keep jet performance unaffected. When $p$ is low, the higher the $\alpha_0$, the lower the system amplitude at the same pressure. When $p$ is high, the amplitude of the jet system is almost unchanged regardless of the changing gas content.

It can also be seen from Figure 10 that when $p$ is greater than 1.2 MPa, the amplitude of the jet system changes smoothly. When $p$ is less than 1.2 MPa, the amplitude of the system decreases in the form of step, and the amplitude at this stage is more sensitive to the change of $p$.

In summary, when the four parameters $\alpha_0$, $p$, $\omega$, and $F_0$, are within the design range, the motion of the jet system is a typical single-cycle vibration with no multi-cycle, bifurcation, and chaos, indicating that the vibration of the jet system of the designed adaptive fire-fighting monitor is regular and predictable in the current parameter range. When $\omega$ is in a certain interval, the jet system has amplitude mutation, and the amplitude of the jet system near the mutation is large. Therefore, in the design of the fire-fighting jet system, the input shaft speed of the pump and the pulsation frequency of the output fluid should avoid the interval. Compared with the wheelset system with obvious bifurcation dynamics in [18], the motion of the adaptive fire-fighting jet system in this paper is single-cycle, when within the range of the design parameters, indicating that the design of the jet system was reasonable.

## 4. Sensitivity Analysis of the Jet System

The steady-state amplitude of the jet system is mainly related to $\alpha_0$, $\omega$, and $p$. In order to analyze the influence of the three variables on the steady-state amplitude, the sensitivity of the steady-state amplitude to the three variables is calculated by the difference method. Due to the interactions between the three variables, the sensitivity of the jet system is analyzed according to the conditions in Section 3.3, Section 3.4, and Section 3.5.

When $p$ is 0.6 MPa and $F_0$ is 3 N, the sensitivity variation curves of the steady-state amplitude $y$ with respect to $\omega$ and $\alpha_0$ are shown in Figures 11 and 12, respectively.

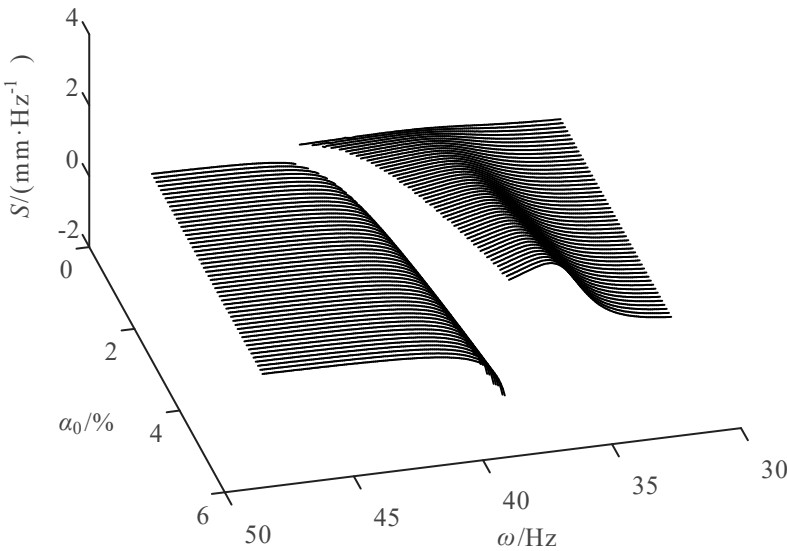

**Figure 11.** Sensitivity variation curve of the steady-state amplitude with respect to $\omega$.

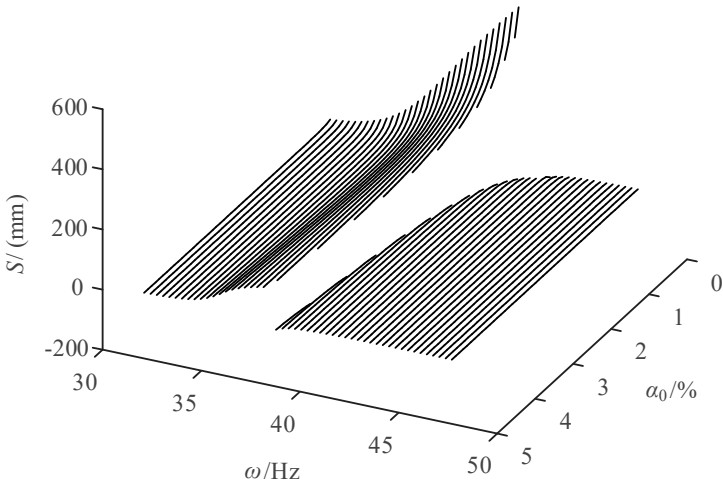

**Figure 12.** Sensitivity variation curve of the steady-state amplitude with respect to $\alpha_0$.

It can be seen from Figure 11 that in the low-frequency range before the amplitude mutation, the sensitivity of the amplitude to $\omega$ is positive. When $\alpha_0$ is constant, the sensitivity increases first and then decreases gradually as $\omega$ increases. When $\alpha_0$ increases, the sensitivity under the same frequency also increases gradually. Meanwhile, the larger the $\alpha_0$, the shorter the frequency range corresponding to the sensitivity variation curve of the low frequency and the greater the maximum sensitivity. In the high-frequency range after the amplitude mutation, the sensitivity of the amplitude to $\omega$ is negative. When $\alpha_0$ is constant, the sensitivity gradually increases and approaches zero as the frequency increases. When $\alpha_0$ increases gradually, the sensitivity slightly increases. Besides, the larger the $\alpha_0$, the longer the frequency range, corresponding to the sensitivity variation curve of the high frequency and the smaller the minimum sensitivity.

It can be seen from Figure 12, that in the low-frequency range before the amplitude mutation, the sensitivity of the amplitude to $\alpha_0$ is positive. When $\omega$ is constant, the sensitivity decreases first and then increases as $\alpha_0$ increases. When $\omega$ increases, the sensitivity under the same gas content also increases gradually. Meanwhile, the higher the $\omega$, the shorter the gas-content range corresponding to the sensitivity variation curve of the low frequency and the greater the maximum sensitivity. In the

high-frequency range after the amplitude mutation, the sensitivity of the amplitude to $\alpha_0$ is negative. When $\omega$ is constant, the sensitivity gradually increases and approaches zero as $\alpha_0$ increases. When $\omega$ gradually increases, the sensitivity under the same gas content also increases. Besides, when the frequency is relatively low, the lower the $\omega$, the shorter the gas-content range, corresponding to the sensitivity variation curve of the high frequency and the smaller the maximum sensitivity.

When $\alpha_0$ is 2% and $F_0$ is 3 N, the sensitivity variation curves of the steady-state amplitude $y$ with respect to $p$ and $\omega$ are shown in Figures 13 and 14, respectively.

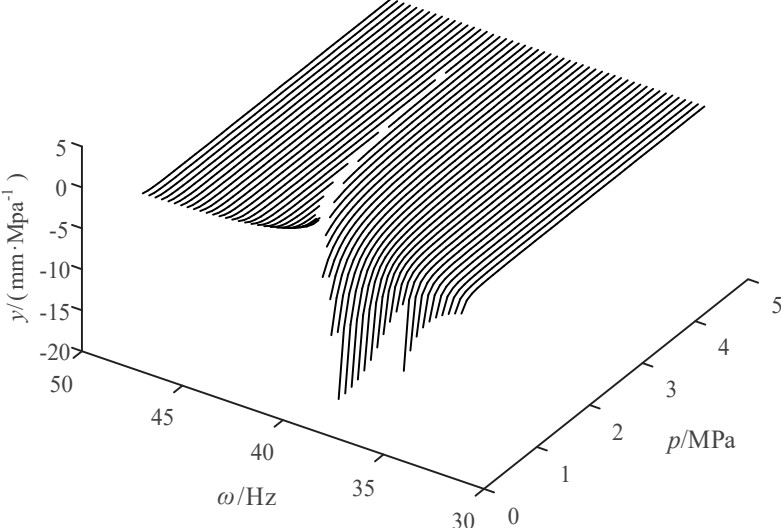

**Figure 13.** Sensitivity variation curve of the steady-state amplitude with respect to $p$.

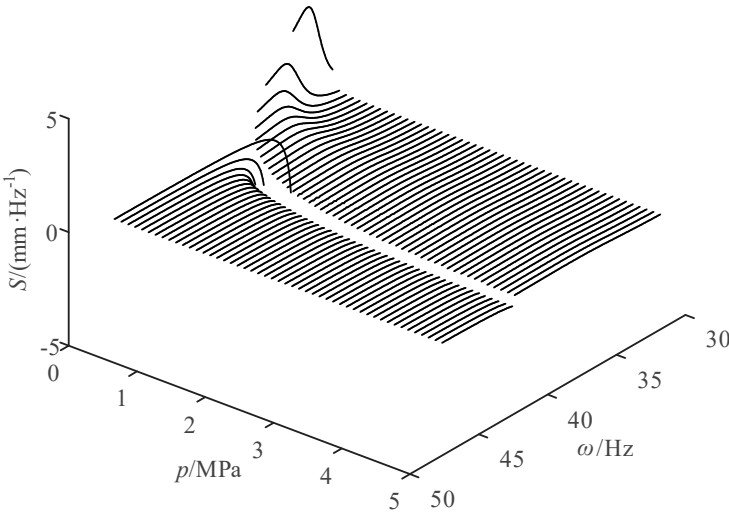

**Figure 14.** Sensitivity variation curve of the steady-state amplitude with respect to $\omega$.

It can be seen from Figure 13 that in the low-frequency range before the amplitude mutation, the sensitivity of the amplitude to $p$ is negative. When $\omega$ is constant, the sensitivity gradually increases and approaches zero, as $p$ increases. When $\omega$ gradually increases, the sensitivity shows a stepwise distribution in the low-pressure region, which corresponds to the amplitude-variation tendency under low frequency and pressure, as shown in Figure 9. In the high-pressure region, the sensitivity gradually increases and approaches zero. In the high-frequency range after the amplitude mutation, the sensitivity of the amplitude to $p$ is positive. When $\omega$ is constant, the sensitivity gradually decreases and approaches zero as $p$ increases. When $\omega$ gradually increases, the sensitivity under the same pressure gradually decreases. Besides, when the frequency is relatively low, the shorter the pressure

variation range, corresponding to the sensitivity variation curve of the high frequency and the larger the maximum sensitivity.

It can be seen from Figure 14 that in the low-frequency range before the amplitude mutation, the sensitivity of the amplitude to $\omega$ is positive. When $p$ is constant, the sensitivity increases first and then decreases as $\omega$ increases. When $p$ gradually increases, the sensitivity under the same frequency gradually decreases. Meanwhile, the larger the $p$, the longer the frequency range, corresponding to the sensitivity variation curve of the low frequency and the smaller the maximum sensitivity. In the high-frequency range after the amplitude mutation, the sensitivity of the amplitude to $\omega$ is negative. When $p$ is constant, the sensitivity gradually increases and approaches zero as $\omega$ increases. When $p$ gradually increases, the sensitivity under the same frequency gradually decreases. Besides, the larger the $p$, the shorter the frequency range, corresponding to the sensitivity variation curve of the high frequency and the smaller the maximum sensitivity.

When $\omega$ is 46.8 Hz and $F_0$ is 3 N, the sensitivity variation curves of the steady-state amplitude $y$ with respect to $p$ and $\alpha_0$ are shown in Figures 15 and 16, respectively.

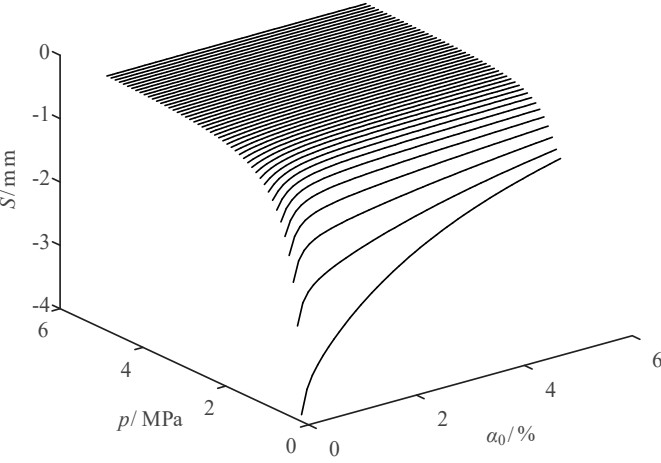

**Figure 15.** Sensitivity variation curve of the steady-state amplitude with respect to $\alpha_0$.

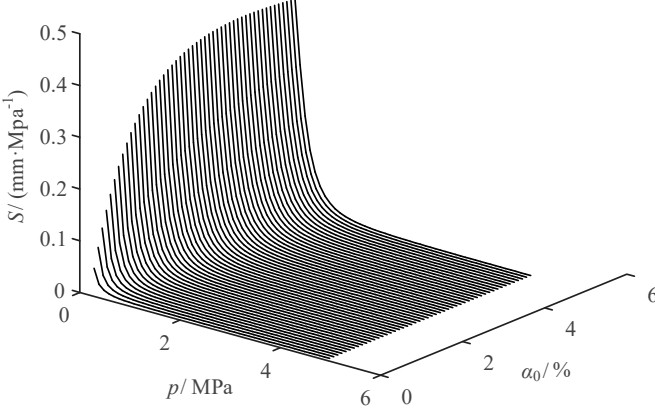

**Figure 16.** Sensitivity variation curve of the steady-state amplitude with respect to $p$.

It can be seen from Figure 15 that the sensitivity of the amplitude to $\alpha_0$ is negative. When $p$ is constant, the sensitivity gradually increases with the increase of $\alpha_0$. Meanwhile, the lower the $p$, the greater the change of the sensitivity variation curve. When $p$ gradually increases, the sensitivity under the same gas content gradually increases.

It can be seen from Figure 16 that the sensitivity of the amplitude to $p$ is positive. When $\alpha_0$ is constant, the sensitivity gradually decreases as $p$ increases. Besides, the larger the $\alpha_0$, the greater the

change of the sensitivity variation curve. When $\alpha_0$ gradually increases, the sensitivity under the same pressure gradually increases.

From the above sensitivity analysis, it is known that when the external excitation frequency variation range includes the mutation frequency, the sensitivity curve will mutate, i.e., a positive sensitivity value changes to be a negative one or a negative value becomes a positive one, which corresponds to the mutation in amplitude variation curve.

It can be seen from Figures 11–16 that within the normal working range, the sensitivity of the jet system amplitude to $\alpha_0$ ranges from −43.62 mm to 519 mm, the sensitivity to $\omega$ ranges from −3.34 mm/Hz to 4.39 mm/Hz, and the sensitivity to $p$ ranges from −18.03 mm/MPa to 3.543 mm/MPa. Since the dimensions of the three variables are different, we cannot compare the sensitivity of the amplitude with respect to different factors. When the design parameters of the jet system vary within a given range, the degree of influence of the parameters on the amplitude can be analyzed according to the sensitivity variation under different parameters.

## 5. Dynamic Experiment of the Jet System

Dynamic research methods are used in this section to analyze the dynamic data of the jet system of the fire-fighting monitor, and to verify the rationality of the dynamic performance of the designed adaptive fire-fighting monitor.

### 5.1. Composition of the Experimental System

According to Figure 17, a platform for the dynamic experiment of the jet system of the fire-fighting monitor was built, which could collect data of the flow, pressure and acceleration of the jet system under different working conditions. During the experiment, the throttle valve was fully opened, and the diesel was used to adjust the system flow. Signals including the pressure signal at the entrance of the gun head were collected by the pressure sensor, and the acceleration signal of the enclosure the gun head collected by the acceleration sensor were transmitted to the DAQ card, and then the computer for data processing and display. The pressure sensor 10 was a MIK-P300 acceleration sensor from MEACON, the acceleration sensor 11 was a 603C01 single-axis acceleration sensor from PCB, and the data acquisition card 12 was a spider20E four-channel dynamic signal analyzer from Crystal Instruments. The fire-fighting monitor prototype and sensors are shown in Figure 18. The acceleration sensor 2 in Figure 18 could collect acceleration data in three directions, which could be used to evaluate the axial vibration state of the jet system. The acceleration sensor 2 was the 356a24 three-axis acceleration sensor of PCB Piezotronics, Inc.

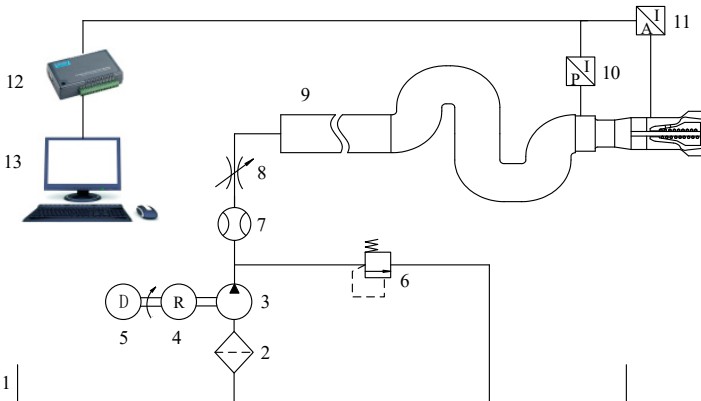

**Figure 17.** Experimental system for the dynamic test of the jet system. The labels are as follows: 1. Water tank, 2. Filter, 3. Pump, 4. Reducer, 5. Diesel, 6. Relief valve, 7. Flowmeter, 8. Throttle valve, 9. Adaptive fire-fighting monitor, 10. Pressure sensor, 11. Acceleration sensor, 12. Data Acquisition card, and 13. Computer.

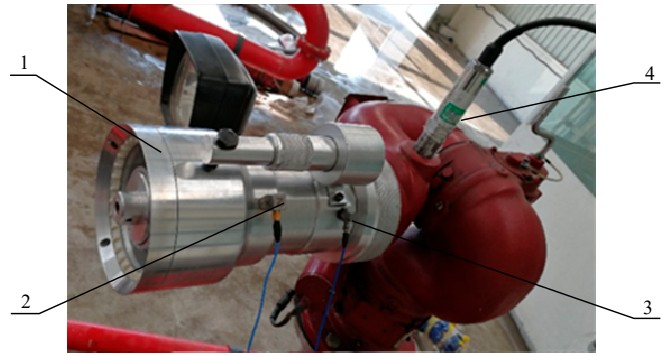

**Figure 18.** The fire-fighting monitor prototype and sensors. The labels are as follows: 1. Fire-fighting monitor 2. Acceleration sensor (three directions), 3. Acceleration sensor (one direction), and 4. Pressure sensor.

*5.2. Acquisition and Analysis of Signals*

5.2.1. Acquisition and Analysis of the Pressure Signal

Under the normal working condition of the jet system, the dynamic test of the fire-fighting monitor was carried out under different flow shown in Table 2. With the recorded data of the pressure at the entrance of the gun head under the corresponding flow, the average pressure $P$, the fluctuation value $\Delta P$, and the load fluctuation value $\Delta F$ of the entrance of the gun head were obtained by calculation.

**Table 2.** State parameters of the entrance of the gun head under different flow.

| Flow/(L/s) | $P$/(MPa) | $\Delta P$/(KPa) | $\Delta F$/(N) |
|:---:|:---:|:---:|:---:|
| 40 | 0.59 | 0.68 | 4.23 |
| 50 | 0.63 | 0.62 | 3.86 |
| 60 | 0.71 | 0.6 | 3.73 |

It can be seen from Table 2, that as the flow of the jet system increase, the pressure at the entrance of the gun head gradually increases, but the fluctuation range tends to gradually decrease, which is because the damping of the jet system becomes larger as the load pressure increases.

5.2.2. Acquisition and Analysis of the Acceleration Signal

The acceleration sensor was used to collect the axial vibration signal during the operation of the adaptive fire-fighting monitor. The data processing methods such as zero-mean processing, wavelet denoising and frequency domain integration were used to preprocess the acquired signal. Then data analysis was carried out by time course, stroboscopic sampling, and power spectrum methods, which are commonly used in nonlinear dynamics.

The relationship between the axial displacement signal and time when the flow of the jet system was 40 L/s, 50 L/s, and 60 L/s is shown in Figure 19. It was found that with the increase of the flow, the amplitude of the axial vibration of the measuring position decreased slightly, but the change was not obvious. The change of the amplitude of the vibration was mainly caused by the change of the pressure.

The power spectrum of the jet system at flows of 40 L/s, 50 L/s, and 60 L/s is shown in Figure 20. It can be seen in the figure that the power spectrum under different working conditions fluctuated gently, indicating that the fluid spring had no obvious alternating transformation between the soft one and the hard one, and there was no cavitation inside the fire-fighting monitor, or the possibility of cavitation was low.

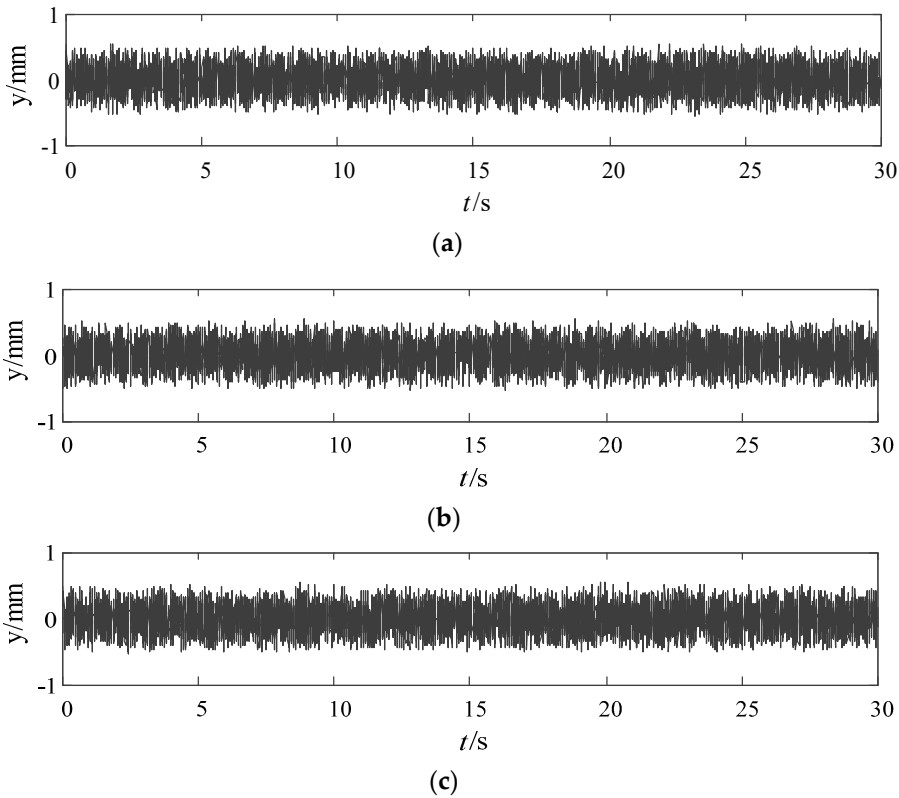

**Figure 19.** Time-domain waveform of vibration displacement signal under different flow. (**a**) Flow of 40 L/s, (**b**) Flow of 50 L/s, (**c**) Flow of 60 L/s.

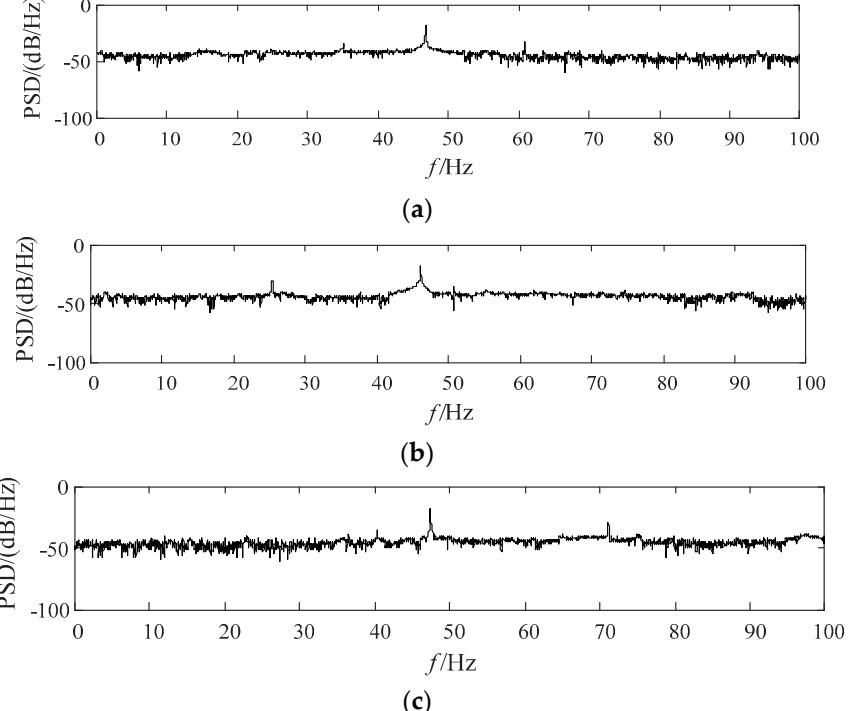

**Figure 20.** Power spectrum of the jet system under different flows. (**a**) Flow of 40 L/s, (**b**) Flow of 50 L/s, (**c**) Flow of 60 L/s.

The stroboscopic sampling of the jet system at flows of 40 L/s, 50 L/s, and 60 L/s are shown in Figure 21. It can be seen that the stroboscopic sampling of the jet system under different flows is

gathered in the fixed area rather than dispersedly distributed in the whole plane, and all the points on the stroboscopic sampling pattern form an oblique elliptical shape with limit-cycle oscillation, indicating that the designed adaptive fire-fighting monitor had no multi-cycle, bifurcation, or chaos under the corresponding design parameters and external excitation. Besides, the single-cycle characteristic result achieved with the experiment is consistent with the simulation results.

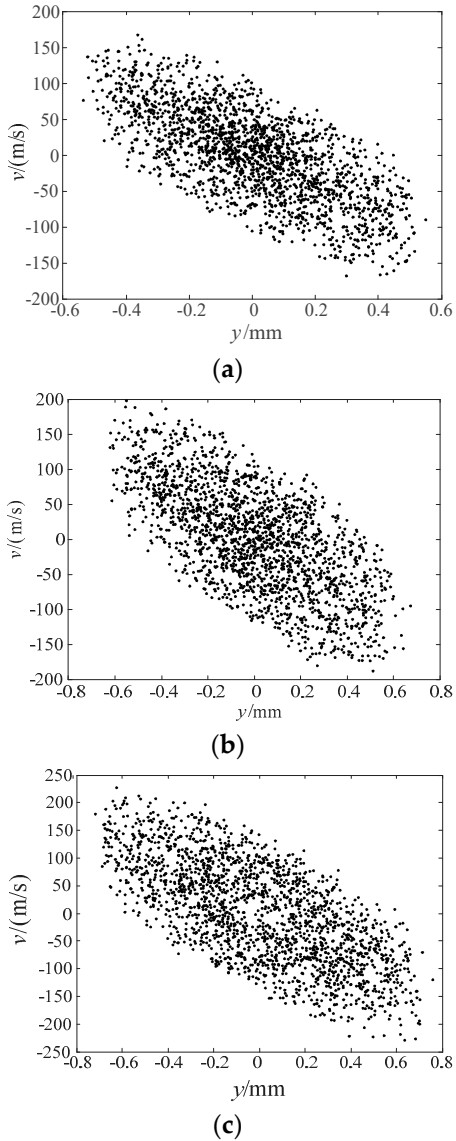

**Figure 21.** Stroboscopic sampling of the jet system under different flows. (**a**) Flow of 40 L/s, (**b**) Flow of 50 L/s, (**c**) Flow of 60 L/s.

## 6. Conclusions

(1) During the operation of the adaptive fire-fighting monitor, the fluid spring stiffness changes dynamically with the gas content and pressure of the jet system, and the nonlinear effect of the fluid spring stiffness can be described by the Duffing equation with damping.

(2) Compared with the dynamic system composed of linear spring and linear damping, the soft spring characteristic of the fluid reduces the vibration amplitude of the jet system of the adaptive fire-fighting monitor at the equilibrium position, which to some extent, can weaken the vibration tendency of the spray core. When the external excitation frequency continuously changes, amplitude mutation occurs near the natural frequency of the corresponding linear system, and the amplitude of

the jet system is large near the mutation frequency. Therefore, in the design of a fire-fighting jet system, the input shaft speed of the pump and the pulsation frequency of the output fluid should avoid the interval where the mutation happens.

(3) Under the corresponding design parameters and external excitation, the designed adaptive fire-fighting monitor always maintained single-cycle motion without multi-cycle, bifurcation, or chaos, which was consistent with the stroboscopic sampling results of the dynamics experiment, verifying the rationality of the design of the adaptive fire-fighting monitor.

**Author Contributions:** Conceptualization, X.Y. and C.W.; Methodology, X.Z.; Investigation, X.Z. and C.W.; Writing-Original Draft Preparation, X.Y.; Writing-Review & Editing, C.W. and X.Z.; Supervision, L.Z. and Y.Z.

**Funding:** This work was funded by the National Natural Science Foundation of China (No. 51805468, 51805214), the Natural Science Foundation of Hebei Province (No. E2017203129), the Open Foundation of the State Key Laboratory of Fluid Power and Mechatronic Systems (No. GZKF-201820), the Basic Research Special Funding Project of Yanshan University (No. 16LGB001), and the China Postdoctoral Science Foundation (No. 2019M651722). The authors would also like to thank the reviewers for their valuable suggestions and comments.

**Conflicts of Interest:** The authors declare no conflict of interest.

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
