# Peer review of "Research on the Dynamic Behaviors of the Jet System of Adaptive Fire-Fighting Monitors"

_processes, doi:10.3390/pr7120952_

Round 1
Reviewer 1 Report
In general, three different values for a parameter are considered to study its effect on a quantity of interest, such as the amplitude. Therefore, consider three different values of β to study its impact on y1. Also, is there any reason to choose only one non-zero value of β
In section 3.3, discuss the effect of the gas content of the fluid (α0) on the maximum amplitude.
In section 3.4, discuss the effect of fluid pressure (p) on the maximum amplitude.
Is there any relation between the state parameters used in sections 3 and 4. It is recommended to use the same symbol to represent a state variable.
Is the statement in line 317 complete?
Reviewer 2 Report
The article combines theoretical model, numerical simulation and experimental validation of the dynamic behavior of jet system of an adaptive nozzle. The model developed by authors conformed to the observed experimental behavior of the experimental system. The develop model is useful beyond the fire-fighting monitors in predicting behaviors of other adaptive nozzles such as non-combustion, i.e. pressurized rocket engines and manouvering water jets actuators. The reviewer suggests for authors to describe in a more details the simulation code, and to post simulation code together with manuscript, as a supplemental info. Authors should also describe in a greater details the experimental setup, and should list the origin/manufacturers of all of the components used in experiment. Authors should also describe their DAC system requirements. Another improvement will be to give an example in introduction of an analogous systems, like for example soft gimbal suspended reactive nozzle, for example in pressure-driven rocket engines, and for jet manouvering trusters, etc... As such, the manuscript deserves to be published after minor modifications as per above.
Reviewer 3 Report
After reading the article entitled "Research on Dynamic Behaviors of Jet System of 3 Adaptive Fire-fighting Monitor", I met good start in defining the problem, but results and comparisons need improvement.
Authors should describe precisely that how they cancelled out multi-cycle, bifurcation and chaos. Which parameters led to this phenomenon? and why mathematically it happened?
There should be comparison with other Dynamical systems which have the same problems like Bifurcation and Chaos and mention what they did to prevent their system from those destructive effects. Authors can find some examples for those effects such as - Effects of the bogie and body inertia on the nonlinear wheel-set hunting recognized by the hopf bifurcation theory.
There could be a sensitivity analysis added to the paper to show the most effective and least one to their system in having single-cycle motion.
Round 2
Reviewer 3 Report
-Authors could answer first question and should add it to the manuscript for readers to clarify the fact.
-Unfortunately, authors did not bring any graph for sensitivity analysis and they did not compare their system to the suggested Dynamical system in previous comments.
-Authors should add their answers in the manuscript for more clarification.
-There should be comparison with other Dynamical systems which have the same problems like Bifurcation and Chaos and mention what they did to prevent their system from those destructive effects. Authors could compare their system for those effects to the following Dynamical system - Effects of the bogie and body inertia on the nonlinear wheel-set hunting recognized by the hopf bifurcation theory.
-There could be a sensitivity analysis added to the paper (using graphs) to show the most effective and least one to their system in having single-cycle motion.
